# Air-Drying Time Affects Mortality of Pyrethroid-Susceptible *Aedes aegypti* Exposed to Transfluthrin-Treated Filter Papers

**DOI:** 10.3390/insects15080616

**Published:** 2024-08-15

**Authors:** Dae-Yun Kim, Jeffrey Hii, Theeraphap Chareonviriyaphap

**Affiliations:** 1Department of Entomology, Faculty of Agriculture, Kasetsart University, Bangkok 10900, Thailand; daeyun.kim@ku.ac.th; 2Research and Lifelong Learning Center for Urban and Environmental Entomology, Institute for Advanced Studies, Kasetsart University, Bangkok 10900, Thailand; 3College of Public Health, Medical & Veterinary Sciences, James Cook University, Brisbane, QL 4000, Australia; hiijk1@gmail.com

**Keywords:** transfluthrin, *Aedes aegypti*, high-throughput screening system, air-drying time, toxicity bioassay

## Abstract

**Simple Summary:**

This study emphasizes the significance of appropriate air-drying times in toxicity bioassays to accurately establish sublethal concentrations and discriminating concentrations for resistance detection in mosquitoes to a highly volatile pyrethroid, namely, transfluthrin. The high-throughput screening system toxicity bioassay study demonstrated consistent dose-dependent responses in susceptible mosquito populations. Our findings emphasize the importance of accurate susceptibility testing to facilitate early resistance detection. The air-drying duration significantly affected the efficacy of transfluthrin: after drying for 24 h, the concentration needed to achieve the same level of efficacy was 2.8 times higher compared to the concentration needed after 1 h of drying. This is the first study to evaluate spatial repellents using a high-throughput screening system toxicity bioassay, yielding precise sublethal concentrations and discriminating concentrations for varying air-drying times of transfluthrin-treated filter papers. The study underscores the importance of early detection of resistant mosquito populations and emphasizes the need to optimize air-drying durations in toxicity bioassays. Selecting the right concentrations and assessing behavioral responses are crucial for developing effective mosquito control strategies with spatial repellents. This research enhances our understanding of resistance dynamics and provides guidance for practical implementation in vector control programs.

**Abstract:**

Increasing temperature can enhance the geographical spread and behavior of disease vector mosquitoes, exposing vulnerable populations to *Aedes*-borne viruses and infections. To address this risk, cost-effective and sustained intervention vector control tools are required, such as volatile pyrethroid spatial repellents. This study used a high-throughput screening system toxicity bioassay to determine the discriminating concentrations of transfluthrin-treated filter papers with variable air-drying times exposed to pyrethroid-susceptible *Aedes aegypti* mosquitoes. At the highest transfluthrin concentration (0.01706%), a significant reduction in mosquito mortality was observed in filter papers air-dried for 24 h compared to those air-dried for 1 h (odds ratio = 0.390, *p* < 0.001, 95% confidence interval: 0.23–0.66). Conversely, no significant difference in mortality was found between filter papers air-dried for 1 h and those air-dried for 12 h (odds ratio = 0.646, *p* = 0.107, 95% confidence interval: 0.38–1.10). The discriminating concentration was 2.8-fold higher for transfluthrin-treated filter papers air-dried for 24 h than it was for papers air-dried for 1 h, and it increased 5-fold from 1 h to 336 h of air-drying. These results show that the optimal air-drying period of transfluthrin-treated filter paper is critical, as higher discriminating concentration values may lead to underestimations of insecticide resistance. The instability of transfluthrin-treated papers necessitates the use of the World Health Organization (WHO) bottle bioassay, which is the preferred method for determining mosquito susceptibility to volatile insecticides.

## 1. Introduction

Dengue is a rapidly expanding arbovirus disease causing approximately 400 million infections annually, with 4 billion people at risk in 128 countries [1,2,3]. This global public health threat is largely driven by uncontrolled urbanization, globalization, the lack of effective vector control interventions, and the increasing insecticide resistance of the mosquito vector, *Aedes aegypti* [4,5,6]. Domestication of this mosquito has considerably impacted its vectorial capacity due to its anthropophilic and endophilic types of behavior in domestic environments [7], coupled with a close association with human-made cryptic larval habitats [8,9].

Insecticide-based interventions, a cornerstone in vector control strategies, face considerable challenges due to the rapid emergence and spread of resistance to WHO-approved chemical classes [10], particularly resistance to two commonly used classes of insecticides: pyrethroids and organophosphates [11,12]. Despite considerable investments in the WHO-recommended control measures for larval habitats, such as larvicide, source reduction, and space spraying [5,13,14], these strategies are losing their efficacy against *Ae. aegypti*, necessitating a critical re-evaluation of existing control measures [10]. Pesticide resistance issues have prompted the investigation of alternative approaches [15], including the use of spatial repellents to deter host–vector contact, that may potentially delay the development of resistance [16].

A previous study successfully determined DCs (discriminating concentrations) and LCs (lethal concentrations) for two pyrethroid spatial repellents, namely transfluthrin and metofluthrin, using a high-throughput screening system for toxicity bioassay (HITSS-TOX) [17]. Although that study identified the different chemical properties of these two active ingredients and demonstrated that metofluthrin had a 4.7-fold greater DC than transfluthrin, the investigation omitted an assessment and comparison of their stability and retention time on paper substrates; this omission could lead to misrepresentation of the actual concentrations at the time of testing, because the spatial repellents begin to vaporize soon after application [18]. The current study aimed to fill this gap in the current knowledge by investigating the stability and retention time of volatile pyrethroid spatial repellents (specifically transfluthrin) on paper substrates. The assessment of these factors should provide insights into the potential impact of substrate properties on the efficacy of these repellents, thereby contributing to the optimization of the HITSS-TOX. The multi-functional HITSS device was initially designed to gather more information on two types of mosquito behavioral responses—contact and noncontact irritancy [19]—by observing their movement away from areas treated with sub-lethal concentrations with or without physical contact [19,20,21]. An optimal concentration of the testing chemical is crucial for conducting dose-dependent assays to observe the behavioral response of mosquitoes [22].

Previous research has verified the emergence of transfluthrin resistance in field populations of *Ae. aegypti* using WHO tube [18] and excito-repellency [23] bioassays, which assess contact toxicity and behavioral avoidance, respectively. The authors observed that responses to transfluthrin-treated filter papers were dose- and time-dependent, as the volatile pyrethroid degrades gradually over time. Consequently, the new international guidelines for susceptibility testing recommend the use of sealed glass containers for evaluating highly volatile pyrethroid spatial repellents [24] to address concerns regarding the instability of filter papers [25].

The current study demonstrated the instability of transfluthrin-treated filter papers and compared their toxicity at various air-drying time points using the HITSS-TOX, including the time-dependent knockdown and killing effects (mortality) of transfluthrin against a pyrethroid-susceptible *Ae. aegypti* strain. Understanding the stability and retention time of volatile pyrethroid spatial repellents on paper substrates is crucial for optimizing their effectiveness against mosquitoes.

## 2. Materials and Methods

### 2.1. Mosquitoes

The *Ae. aegypti* laboratory strain USDA was originally sourced from the United States Department of Agriculture in Gainesville, FL, USA. This strain, which is susceptible to insecticides, has been kept under controlled laboratory conditions for more than 20 years at the Department of Entomology, Faculty of Agriculture, Kasetsart University in Bangkok, Thailand.

The immature stages were raised to adulthood under controlled conditions of 27 ± 2 °C, 80 ± 10% relative humidity (% RH), and 12 h light/dark cycle. Upon emergence, adult mosquitoes were given access to cotton pads soaked with 10% sucrose solution and kept in individual insectaries. On the third day after emergence, naturally mated females were allowed to feed on expired human blood obtained from Thai Red Cross Society using glass feeders covered with pig’s intestine. Two days after blood feeding, 10 cm diameter oviposition dishes with moist filter paper were placed in the adult holding cages to encourage egg laying. The eggs were then air-dried at room temperature for 1–2 days to complete embryonic development before being transferred to clean water in individual rearing trays (30 cm long × 20 cm wide × 5 cm high). To ensure uniform mosquito body sizes, each tray, containing approximately 200 larvae, was fed daily with a commercial fish protein mixture (Optimum^TM^ Nishikigoi Carp Fish; Perfect Companion Group Co., Ltd.; Samutprakarn, Thailand).

### 2.2. Chemicals

Technical-grade 97.9% transfluthrin provided by SC Johnson (Racine, WI, USA) was used to prepare the transfluthrin stock solution by mixing with analytical-grade acetone (Avantor Performance Materials, Inc., Allentown, PA, USA) and silicone oil (Dow Corning^®^ 556 cosmetic grade, Dow Chemical Company, Midland, MI, USA and Corning, Inc., Midland, MI, USA) at an acetone-to-silicone oil ratio of 2.05:1.01. Serial dilutions were prepared from the stock solution and were used to impregnate filter papers, each with a surface area of 275 cm^2^. The concentrations used for the HITSS-TOX were 0.00107% (equivalent to 0.393 mg/m^2^), 0.00213% (0.782 mg/m^2^), 0.00427% (1.568 mg/m^2^), 0.00853% (3.133 mg/m^2^), and 0.01706% (6.266 mg/m^2^).

### 2.3. Transfluthrin-Impregnated Filter Papers

Whatman No.1 filter papers (Whatman International Ltd., Banbury, UK), each measuring 11 cm × 25 cm, were treated with the serial concentrations of transfluthrin solutions. A micropipette was used to apply 3.0 mL aliquots of each concentration to each paper. Then, the transfluthrin-treated papers were air-dried on the metal pins of a holding rack for 1 h up to 672 h under laboratory conditions (27 ± 2 °C, 80 ± 10% RH, and a 12-h light/dark cycle with indoor fluorescent lighting). Serial concentrations were prepared to determine the DCs for each air-drying time point. The DCs were determined by doubling the 99% LC (LC_99_) for the susceptible *Ae. aegypti* (USDA) strain [26]. For the control group, filter papers treated with the acetone-silicone oil carrier, excluding transfluthrin, were compared.

### 2.4. High-Throughput Screening System

The HITSS device utilized in this study featured three interconnected cylinders, allowing for three different configurations based on the module setup [19]. The toxicity module, designed to measure lethal concentrations, included a single metal chamber with an end cap and a funnel section. Each chamber was equipped with filter paper treated with a specific concentration of transfluthrin for the treatment group. In contrast, the control group chambers contained filter paper treated with a mixture of acetone and silicone oil but without transfluthrin.

For each replicate, 20 nulliparous, mated, non-blood-fed, female mosquitoes aged 3–5 d were given access to a 10% sucrose solution on a moist cotton wick and were starved for 12 h before testing (during which only water was provided). The mosquitoes were moved into the metal chamber using a mouth aspirator, and the number of mosquitoes showing knockdown was recorded after a 60 min exposure to each concentration. After exposure, the test mosquitoes were transferred to a holding cup containing a cotton ball soaked in 10% sucrose solution. They were then maintained under laboratory conditions (27 ± 2 °C, 80 ± 10% RH, 12 h light/dark cycle) for 24 h. The mortality of the mosquitoes at each concentration was observed and recorded during this period. Six replicates were performed for each of the five concentrations and the control at the seven different air-drying times: 1, 12, 24, 168, 336, 504, and 672 h tested using the TOX. The mosquito recovery rate was calculated by subtracting the mortality observed at 24 h from the knockdown after 1 h of exposure.

### 2.5. Data Analysis

A normality test (the Shapiro–Wilk test) was performed on the data to assess whether they followed a normal distribution. For data that followed a normal distribution, a one-way analysis of variance was conducted, with Tukey’s honestly significant difference test for multiple comparisons and Dunnett’s T3 test for pairwise comparisons. For 2-sample comparisons, Student’s *t*-test was applied with a significance level of *p* = 0.05. When datasets had unequal variances, Welch’s *t*-test was used for comparison.

Log-transformation of the percentage 1 h knockdown and 24 h mortality values was performed to address skewed data and meet the assumptions of statistical models. Either the natural logarithm of *n* or the log 10 (*n* + 1) transformation was used. If the log-transformed data still did not meet the assumption of equal variance, non-parametric tests were employed. Specifically, the Kruskal–Wallis *H* test was used for multiple comparisons, and the Mann–Whitney *U* test was applied for two-sample comparisons on the original non-parametric datasets. The mean rank, minimum and maximum ranges, and pairwise comparisons were used to assess significance among the concentrations for 1 h knockdown and 24 h mortality at a significance level of *p* = 0.05. The mean percentages ± standard errors (SEs) of untransformed data are reported in the tables and figures.

The mortality associated with transfluthrin-treated filter papers at different air-drying times is expressed as the odds ratio (OR) with a 95% confidence interval (95% CI) using statistical parameters estimated by fitting a mixed-effect logistic regression model.

Lethal concentrations of transfluthrin were established using probit analysis based on the mortality data collected from each air-drying time point, specifically considering the number of female mosquitoes that were still alive 24 h after exposure, across the five tested concentrations. Discriminating concentrations were determined from the mortality data using Pearson’s goodness-of-fit test to assess the agreement between the observed and expected distributions. The 95% fiducial limits were calculated using maximum likelihood estimates of parameters and log-probit regression analysis based on the baseline data.

The statistical analyses were performed using SPSS software version 29 (IBM, Armon, NY, USA).

## 3. Results

### 3.1. Dose-Dependent Toxicity Bioassay

Exposure to 1 h air-dried filter papers impregnated with 0.01706% transfluthrin gave a significantly higher knockdown compared to that with 0.00427% (*t* = 9.944, df = 10.0, *p* < 0.001) and 0.00107% (*t* = 13.811, df = 5.284, *p* < 0.001) transfluthrin. Similarly, the highest transfluthrin concentration (0.01706%) showed significantly higher mortality compared to the middle concentration (0.00427%: *t* = 8.916, df = 10.0, *p* < 0.001) and the lowest concentration (0.00107%: *t* = 13.302, df = 5.270, *p* < 0.001). Furthermore, when filter papers treated with transfluthrin and air-dried for 24 h were tested, the dose-dependent responses remained consistent with those observed for 1 h air-dried paper (*p* > 0.05) (Table 1).

### 3.2. Toxicity of Transfluthrin-Impregnated Filter Papers at Different Air-Drying Times

Mosquito knockdown was not affected by different air-drying times, as knockdown at the highest concentration (0.01706%) was not significantly different between 1 h (89.2 ± 6.2%) and 24 h (95.0 ± 3.4%) air-dried TFT-treated filter papers (Mann–Whitney *U* = 23.0, *n*_1_ = *n*_2_ = 6, *p* = 0.392), as shown in Table 1. Similarly, knockdown was not significantly different for 672 h (28 days) air-dried papers compared to 1 h air-dried papers (Mann–Whitney *U* = 9.5, *n*_1_ = *n*_2_ = 6, *p* = 0.164), as shown in Appendix A.

The mortality at the highest concentration was not significantly different between 1 h (69.2 ± 5.1%) and 24 h (46.7 ± 11.7%) air-dried filter papers (*t* = 1.760, df = 10.0, *p* = 0.109), as shown in Table 1. While knockdown remained relatively constant with the air-drying time, 24 h mortality showed a declining trend from 1 h (69.2 ± 5.1%) to 672 h (26.7 ± 8.3%) for 0.01706% transfluthrin-impregnated filter paper (*t* = 4.357, df = 10.0, *p* = 0.001), as shown in Appendix A and Figure 1.

### 3.3. Mosquito Recovery Rates of Transfluthrin-Exposed Aedes aegypti

At the two highest transfluthrin concentrations (0.01706% and 0.00853%), the recovery rate was significantly increased from 1 h (20.0 ± 7.1%) to 24 h (48.3 ± 10.0%) air-dried filter papers for both 0.01706% (*t* = −2.318, df = 10.0, *p* = 0.043) and 0.00853% (*t* = −2.407, df = 10.0, *p* = 0.037), as shown in Table 1. However, the recovery rates were not significantly different at lower transfluthrin concentrations.

Although 24 h air-dried filter papers impregnated with 0.01706% transfluthrin showed dose-dependent recovery between 0.00213% (*t* = 4.945, df = 5.279, *p* = 0.004) and 0.00107% (*t* = 4.847, df = 5.000, *p* = 0.005), this relationship was not apparent from 1 h air-dried papers (Table 1). The dose responses persisted for filter papers air-dried for up to 4 weeks (Appendix A).

Figure 1 shows the trends in the knockdown and mortality using a range of 1 h to 672 h air-dried filter papers impregnated with 0.01706% transfluthrin. The knockdown rates were uniformly maintained for up to 672 h (28 days), with a 13.3% difference between the lowest and highest knockdown values, which was not significantly different from the difference observed for 1 h air-dried paper (1 h: 89.2 ± 6.2% vs. 672 h: 82.5 ± 6.6%, Mann–Whitney *U* = 9.5, *n*_1_ = *n*_2_ = 6, *p* = 0.164). In contrast, the 24 h mortality of 0.01706% transfluthrin-exposed *Ae. aegypti* (USDA) significantly dropped with increased air-drying times, with a 42.5% difference between the lowest and highest 24 h mortality rates (Appendix A, Figure 1, lower graph). Subsequently, the gap between knockdown and mortality at the highest concentration significantly increased from 1 h (20.0 ± 7.1%) to 672 h (55.8 ± 7.0%) air-dried filter papers (*t* = −3.601, df = 10.0, *p* = 0.005), as shown in Appendix A.

A strong correlation was observed between air-drying time and recovery at the highest transfluthrin concentration (0.01706%, R^2^ = 0.58; 95% CI: 35.34–54.37), whilst the lowest transfluthrin concentration (0.00107%) showed a very weak negative correlation (R^2^ = 0.00; 95% CI: −0.67 to 0.22), as shown in Figure 2.

### 3.4. Effect of Air-Drying Time on Aedes aegypti Mortality

At the highest concentration of transfluthrin (0.01706%), a significant decrease in mortality was observed for filter papers air-dried for 24 h compared to those air-dried for 1 h (odds ratio (OR) = 0.390, *p* < 0.001, 95% confidence interval (CI): 0.23–0.66). In contrast, no significant difference in mortality was detected between filter papers air-dried for 1 h and those air-dried for 12 h (OR = 0.646, *p* = 0.107, 95% CI: 0.38–1.10), as detailed in Table 2.

### 3.5. Effect of Air-Drying Time on Discriminating Concentrations

The established 50%, 75%, and 99% lethal concentrations (LC_50_, LC_75_, and LC_99_) of transfluthrin for the 1 h and 24 h air-dried filter papers were significantly different (*p* < 0.05). Consequently, the discriminating concentration (DC) was 2.8-fold higher for the 24 h air-dried transfluthrin filter paper than for the 1 h air-dried paper, and it increased by 5-fold from 1 h to 336 h air-dried papers, as shown in Table 3. Although the 12-h air-dried papers had 1.4-fold higher DC compared to the 1-h air-dried papers, no significant differences in the LC were observed. Furthermore, the chi-square statistic varied with the air-drying time, with the 12-h air-dried paper showing a notably higher value of 10.417. This highlights the importance of air-drying duration in the accuracy of LC estimates.

## 4. Discussion

The main aim of this study was to provide valuable insights into the impact of air-dried filter papers on the stability and effectiveness of transfluthrin, thereby contributing to the refinement of testing methodologies in accordance with the latest WHO guidelines for both contact toxicity [24] and spatial repellency [27].

The current results revealed considerable variability in mosquito mortality rates associated with DCs that have a direct impact on the choice of susceptibility bioassay methods, as a low level of sensitivity was reported in comparisons of populations over time and space [28,29]. For example, the 24 h mortality associated with transfluthrin decreased linearly as the air-drying time of the papers increased from 1 h to 672 h, while the 1 h knockdown rate stayed relatively constant at around 90%. The difference between the mortality and knockdown rates was used to represent mosquito recovery, which doubled from 1 h to 24 h air-drying time. This air-borne effect may result in DC overestimation due to the volatilization of transfluthrin over 24 h; hence, it is necessary to increase the concentrations by 2.8-fold (0.01706%/0.0853%) to attain the mortality corresponding to LC_99_. This can result in resource wastage and, more importantly, the potential selection of mosquito populations with higher levels of resistance.

Similarly, Sukkanon and colleagues [18] highlighted the importance of establishing an appropriate baseline for DC values derived from the WHO tube bioassay (ideally within 1 h of air-drying post treatment with transfluthrin), as it uses the standard cellulose-based treated filter papers followed by air-drying for 24 h, which may result in transfluthrin instability. Although filter papers are used in both the WHO tube bioassay and HITSS-TOX, direct comparisons of the results can be challenging [30] because of the different settings of experimental conditions, such as the various sizes and volumes of testing equipment, cylinders, or chambers, along with room sizes, yielding different results [31]. For example, bioassays on the pyrethroid-susceptible *Ae. aegypti* strain gave 95% knockdown at 0.01706% (6.266 mg/m^2^) and 0.01250% (4.591 mg/m^2^) transfluthrin on treated filter papers that had undergone 24 h air-drying using the standard WHO procedure [26].

Generally, the inherent limitation of most bioassays is that they fail to provide information on the actual concentrations of the active ingredient at the time of testing [18]. As transfluthrin starts to vaporize soon after application, its stability and retention time on paper or any other treated surface decrease with exposure time. Other studies have reported the initial percentage concentration of treated papers or glass bottles, but they were not assayed to determine the loss of the active ingredient prior to testing [18,25].

The current study identified significant variability in 24 h mortality by indirectly comparing changes in the air-drying intervals of the transfluthrin-treated filter papers. Although recent WHO guidelines recommend testing volatile pyrethroid spatial repellents using bottle bioassays [24], we recently reported variations in the chemical properties of transfluthrin and metofluthrin using the HITSS-TOX [17]. For example, transfluthrin produced DC and knockdown values that were 4.7 times greater than those for metofluthrin. Notably, our research showed that prolonged air-drying durations resulted in lower mosquito mortality but a higher knockdown rate for 0.01706% transfluthrin.

Volatile pyrethroids, such as transfluthrin and metofluthrin, act as sodium channel modulators, disrupting sodium uptake in neurons, leading to prolonged depolarization of the cell membrane [32,33]. This depolarization induces hyperexcitation effects, including flightlessness and moribund states that are indicative of knockdown [34], while modification of the sodium channel by pyrethroids can cause sustained abnormal hyperexcitability [35], which manifests as incapacitating yet non-lethal knockdown. As laboratory assays of pyrethroids typically rely on knockdown and mortality as criteria for advancing compounds in the screening process [16], standardized assays are needed to evaluate behavioral responses to volatile pyrethroid spatial repellents. Achee and colleagues [36] confirmed that even at low concentrations, certain pyrethroids reduce mosquito entry into experimental huts, with other associated behaviors being affected by volatilization due to the air-drying period, adhesion to the substrate, sublethal doses, and environmental conditions [37].

Jansma and Linders [38] investigated the relationship between air concentrations of the active ingredient and repellency behavior and confirmed that the concentration of 0.00625% metofluthrin used in coils was below the threshold needed to elicit toxicity (mortality) in mosquitoes. However, even at these low concentrations, *Ae. aegypti* entry into experimental huts was reduced by 58% due to variations in the airborne spatial repellent concentrations that were influenced by the height of the coils from the floor, time of day, and ambient temperature [39], as well as by the distance from the volatile pyrethroid spatial repellent emanator [40]. We also showed a proportional correlation between the air-drying period and 24 h recovery rate, with a similar relationship reported between a decreased transfluthrin concentration or sublethal dose of transfluthrin and mosquitoes’ behavioral responses [16,36].

Understanding the importance of lowered insecticide susceptibility is crucial, as the LCs of volatile pyrethroid spatial repellents may mitigate selective pressures that cause resistance to develop in the context of an integrated vector management strategy. The continued use of residual insecticides that utilize the same mechanisms of toxicity at higher doses in both public health and agriculture can result in selection for resistance traits [37]. Similarly, Wagman and colleagues [41] demonstrated the effects of consistent spatial repellency and the high degree of variability in mosquito behavior, with transfluthrin-exposed *Ae. aegypti* not being repelled again at 24 h after their recovery in an HITSS spatial repellent assay. An important factor is the role of physiological drivers of the spatial repellent behaviors caused by volatile pyrethroid spatial repellents and their impact on insecticide resistance in the target vector.

## 5. Conclusions

In conclusion, our study confirmed the critical role of air-drying times in determining the DCs for *Ae. aegypti* using transfluthrin-treated filter papers in an HITSS-TOX assay. Given that different active ingredients have unique baseline LC values and DCs, it is necessary to establish comparable DCs as an initial reference for resistance evaluation in different mosquito populations. Despite the instability of filter papers, our study demonstrated the advantages of calibrating air-drying times and highlighted the implications for understanding the importance of applying highly volatile spatial repellents.

## Figures and Tables

**Figure 1 insects-15-00616-f001:**
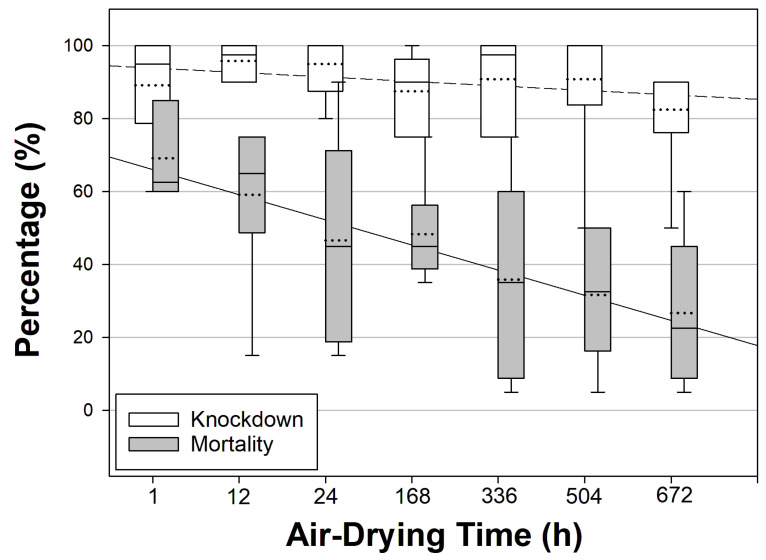
Boxplot displaying the knockdown and mortality of pyrethroid-susceptible *Aedes aegypti* (USDA) exposed to 0.01706% transfluthrin on filter papers dried for 1 to 672 h. The bottom and top lines of the boxes are the 25th and 75th percentiles, with the middle lines marking the medians. Whiskers show the 10th and 90th percentiles, indicating the 95% confidence intervals. The dotted line in each box represents the mean.

**Figure 2 insects-15-00616-f002:**
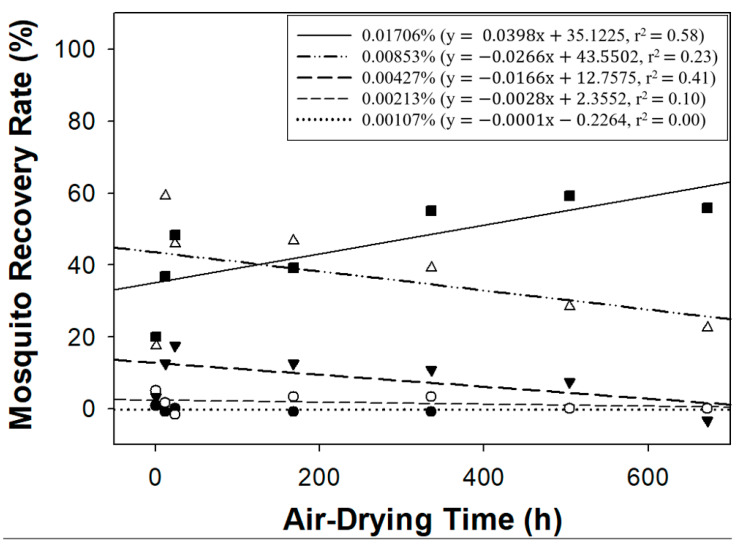
Relationship between recovery rate of pyrethroid-susceptible *Aedes aegypti* USDA strain (%) and air-drying time (hour) of filter papers at five transfluthrin concentrations. Black square = 0.01706%, white triangle = 0.00853%, black triangle = 0.00427%, white circle = 0.00213%, and black circle = 0.00107% transfluthrin.

**Table 1 insects-15-00616-t001:** Percentage knockdown after 1 h, mortality after 24 h, and recovery of *Aedes aegypti* (USDA) female mosquitoes were measured. Mosquitoes were exposed to five concentrations of transfluthrin on filter papers with 1 h and 24 h of air-drying using HITSS-TOX.

TFT Conc. (%)	Mean % (SE) *Ae. aegypti* (USDA) at Air-Drying Time (h)								
	Knockdown		*p*-Value ^†^	Mortality		*p*-Value ^†^	Recovery		*p*-Value ^†^
	1 h	24 h		1 h	24 h		1 h	24 h	
0.01706	89.2 (6.2) a	95.0 (3.4) a	0.392	69.2 (5.1) a	46.7 (11.7) a	0.169	20.0 (7.1) a	48.3 (10.0) a	0.041
0.00853	63.3 (8.4) ab	69.2 (9.9) a	0.809	45.8 (12.7) ab	23.3 (7.1) a	0.153	17.5 (8.3) a	45.8 (8.3) a	0.037
0.00427	17.5 (3.6) b	23.3 (3.3) b	0.262	14.2 (3.5) b	5.8 (2.4) ab	0.078	3.3 (4.8) a	17.5 (5.3) ab	0.075
0.00213	6.7 (2.5) bc	1.7 (1.7) c	0.105	1.7 (1.1) b	3.3 (2.1) b	0.702	5.0 (2.9) a	−1.7 (1.7) b	0.073
0.00107	1.7 (1.1) c	0.0 (0.0) c	0.138	0.8 (0.8) b	0.0 (0.0) b	0.317	0.8 (0.8) a	0.0 (0.0) b	0.317

N = 120 for each time point and transfluthrin concentration (20 females per replicate). Values in the same column with the same lowercase letter are not significantly different. ^†^ Significance between 1 h and 24 h air-dried filter papers was determined using Student’s *t*-test or the Mann–Whitney *U* test (*p* < 0.05). Recovery rate was calculated as %knockdown–%mortality at 24 h. Controls resulted in no knockdown or mortality. USDA: United States Department of Agriculture; TFT: Transfluthrin.

**Table 2 insects-15-00616-t002:** Effect of air-drying time of 0.01706% transfluthrin-treated filter papers on 1 h knockdown and 24 h mortality against pyrethroid-susceptible *Aedes aegypti* (USDA) using HITSS-TOX.

Air-Drying Time (h)	Mean % (SE) Knockdown [95% CI]	Mean % (SE) Mortality [95% CI]	Mortality OR [95% CI]	Z-Value	*p*-Value
1	89.17 (6.25) [73.11–105.23]	69.17 (5.07) [56.14–82.20]	-	63.94	-
12	95.83 (2.01) [90.67–100.99]	59.17 (9.26) [35.37–82.96]	0.646 [0.380–1.099]	2.60	0.107
24	95.00 (3.42) [86.22–103.78]	46.67 (11.74) [16.49–76.84]	0.390 [0.230–0.661]	12.22	<0.001
168	87.50 (4.43) [76.12–98.88]	48.33 (5.73) [33.62–63.05]	0.417 [0.246–0.707]	10.56	0.001
336	90.83 (5.07) [77.80–103.86]	35.83 (10.83) [7.99–63.68]	0.249 [0.145–0.426]	25.67	<0.001
504	90.83 (8.21) [69.74–111.93]	31.67 (7.15) [13.29–50.04]	0.207 [0.120–0.357]	32.06	<0.001
672	82.50 (6.55) [65.66–99.34]	26.67 (8.33) [5.25–48.09]	0.162 [0.093–0.284]	40.53	<0.001

Statistical parameters estimated by fitting mixed-effect logistic regression model (mortality). Mortality OR of filter papers for different air-drying times from 12 to 672 h, compared to mortality at 1 h. OR: odds ratio; CI: confidence interval. Controls resulted in no knockdown or mortality.

**Table 3 insects-15-00616-t003:** Summary of lethal (LCs) and discriminating concentrations (DCs) of a laboratory strain of *Aedes aegypti* (USDA) exposed to five serial transfluthrin concentrations on filter papers at seven air-drying times (hours) using HITSS-TOX.

Air-Drying Time (h)	% LC_50_ (95% FL)	% LC_75_ (95% FL)	% LC_99_ (95% FL)	% DCs	χ^2^ (df) *	*p*-Value
1	0.01040 (0.00925–0.01189) a	0.01852 (0.01576–0.02280) a	0.07611 (0.05455–0.11971) a	0.15222	3.217 (3)	0.359
12	0.01515 (0.01039–0.03612) ab	0.02666 (0.01628–0.12592) ab	0.10636 (0.04148–3.15898) ab	0.21272	10.471 (3)	0.015
24	0.01870 (0.01543–0.02434) b	0.03784 (0.02831–0.05797) b	0.21269 (0.12005–0.50602) b	0.42538	2.468 (3)	0.481
168	0.01790 (0.01480–0.02314) b	0.03711 (0.02784–0.05621) b	0.22121 (0.12524–0.51640) b	0.44242	6.354 (3)	0.096
336	0.02577 (0.01984–0.03820) bc	0.05672 (0.03826–0.10560) b	0.39147 (0.18471–1.31841) b	0.78294	6.848 (3)	0.077
504	0.02742 (0.02083–0.04220) bc	0.05917 (0.03913–0.11677) b	0.38927 (0.17758–1.45757) b	0.77854	3.583 (3)	0.310
672	0.03118 (0.02319–0.05157) c	0.06318 (0.04072–0.13760) b	0.35637 (0.15747–1.56318) b	0.71273	1.301 (3)	0.729

N = 120 for each time point and concentration (20 per replicate). * Chi-square goodness-of-fit test statistic. The same letters indicate no significant differences in a column based on the FL range. Transfluthrin-treated filter papers were air-dried under room conditions (27 ± 2 °C, 80 ± 10% RH). LC: lethal concentration; FL: fiducial limit; DCs: two-fold probit-derived LC_99_; df: degrees of freedom; TOX: toxicity assay in which the tested mosquitoes were exposed to transfluthrin for an hour.

## Data Availability

The datasets supporting the conclusions of this article are included within the article. Raw data are available from the corresponding author upon reasonable request or can be downloaded via https://zenodo.org/doi/10.5281/zenodo.13253083 (accessed on 14 August 2024).

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
