# Peer review of "Air-Drying Time Affects Mortality of Pyrethroid-Susceptible Aedes aegypti Exposed to Transfluthrin-Treated Filter Papers"

_insects, 2024, doi:10.3390/insects15080616_

Round 1

Reviewer 1 Report

Comments and Suggestions for Authors

General Comments

This study highlights the importance of appropriate air-drying times in toxicity bioassays for accurately determining sublethal and discriminating concentrations in resistance detection.

The paper is challenging to read in its current form, especially the results section. In addition, the paper uses numerous abbreviations, which can be confusing for the reader. It would be helpful to review and ensure that all abbreviations are necessary for clarity and conciseness. I would recommend reducing the number of abbreviations used in the main text by half, at least, to improve readability.

The authors appear to conflate the distinct concepts of repellents, pesticides, and insecticides sometimes. It is important to clearly differentiate between these terms to avoid confusion and ensure accurate communication of the study's findings.

The authors seem to have conflated the discussion and conclusions sections. Conclusions should be concise and summarize the key findings. The discussion section is where the results should be thoroughly analysed and compared with previous studies.

Specific Comments

Title:

It can be shortened by removing “(Diptera: Culicidae)”

Simple summary:

Line 13, authors should refer that the present study focuses on resistance to insecticides (more specifically pyrethroids)

Lines 17-20, This sentence is not clear “… because a 24 h drying time (0.42538%) resulted in an elevated that was discriminating concentration 2.8 times higher than that of 1 hour (0.15222%).” I believe the authors mean that “after drying for 24 hours, the concentration needed to achieve the same level of efficacy was 2.8 times higher compared to the concentration needed after just 1 hour of drying.” Please rewrite and clarify.

Line 21, The author must consider rewriting the sentence to “This study is the first to evaluate spatial repellents using a high-throughput…”

Abstract:

Line 38, authors should consider modifying the sentence “all lethal concentrations (LCs) and DCs were positively correlated with air-drying times (50% LC (LC50)…”

Line 42, insecticides instead of pesticides to avoid confusion

Line 43, state what WHO stands for

Introduction:

Lines 49-50, rephrase for clarity: “Dengue is a rapidly expanding arboviral disease causing approximately 400 million infections annually, with 4 billion people at risk in 128 countries.”

Line 57, simply write WHO

Line 59, erase the comma: “pyrethoids and organophosphates”

Lines 59-63, rephrase for clarity: “Despite considerable investments in WHO-recommended control measures for larval habitats, such as larvicide, source reduction, and space spraying, these strategies are losing their efficacy against Ae. aegypti, necessitating a critical re-evaluation of existing control measures.”

Line 84, Previous research instead of “Relatively recent research”

Material & Methods:

Section 2.1, authors should say that the strain was obtained in 1996 or that it has been maintained for 20 years to avoid repeating the information

Line 101, erase “is so named as it”

Line 108, 10% sucrose instead of 10% sugar solution

Line 110, specify what type/source of blood was used to feed the mosquitoes. It would be interesting to know if the authors use glass feeders, hemotek system or other, and what kind of membrane?

Lines 118 and 119, erase the sentence “Adults were maintained in screen cages with a 10% sucrose solution before assays.” as this information already appears previously in the text.

Lines 121, 123 and 125, United States of America or USA instead of United States

Lines 142 and 143, erase “(for example, the TOX has a single chamber”

Line 146, “control group, the chamber contained a filter paper treated with a solvent without the AI.” – Clarify, which solvent? Which AI?

Line 153, 10% sucrose instead of sugar

Line 156, at this point I am not sure we already know what are the “seven different air–drying times tested”. If not, please specify here.

Line 163, “was performed” instead of were performed & “to assess if it followed…” instead of they

Line 165, “The Kruskal–Wallis H test was used”

Line 184, “The statistical analyses were performed using SPSS software”

Results:

Legend of Table 1 is very confusing. Please rewrite. Also, use the abbreviation in this case instead of high–throughput screening system toxicity bioassay. No need to add this info again after the Table 1 “USDA: United States Department of Agriculture.”

Add a space between lines 205 and 206 and between lines 250 and 251.

Line 206, times instead of time.

Line 206, once again, the authors are not consistently using abbreviations that were previously introduced in the main text. While it is acceptable and even advisable to minimize the use of abbreviations, the authors should decide whether to use them or not and then revise the main text accordingly for consistency. The same applies to the legend of Figure 1.

Lines 210, 214, 227 and 244, Table with a capital letter instead of table

Figure 1 should be improved in quality, and the legend should be made more concise

Lines 212, 217, 232, 240, 253 and 266, Supplementary instead of supplementary and be more precise, supplementary what? Table? Figure? Dataset?

Line 225, a comma should be added after the parenthesis

Line 237, erase as shown in supplementary 2, and Figure 1 as this information appears again in the end of the sentence.

Line 245, abbreviations appearing again “transfluthrin (TFT) treated filter papers on 1 h knockdown (KD) and 24 h mortality (MT) against pyrethroid–susceptible Aedes aegypti (USDA) using high–throughput screening system toxicity bioassay.”. Please use the abbreviation or the word but be coherent along the manuscript.

Discussion:

Inexistent… I don’t understand the discussion section where authors only state: “Authors should discuss the results and how they can be interpreted from the perspective of previous studies and of the working hypotheses. The findings and their implications should be discussed in the broadest context possible. Future research directions may also be highlighted.”

Conclusions:

Only the last paragraph… the rest is discussion.

Lines 288 and 289, please rephrase for improving clarity “For example, the lethality of TFT declined linearly from 1 h to 672 h of air–drying time, while the knockdown rate remained relatively stable (average 90%).”

Line 301: Please correct “such as various sizes and volumes”

Line 353, aegypti instead of Aegypti

Line 359, erase the abbreviation IVM

I recommend erasing lines 377 and 378

Supplementary Materials: please revise the legend of the figures to make it more concise and clearer. Avoid repeating information.

Author contributions:

Line 394, “Graphical Abstract: D.-Y.K”. Where is the graphical abstract?

Data availability:

Raw data should be open and published in Zenodo or similar. Please upload and provide a public code or link for the readers. This facilitates the access to the data and avoids continuous emails asking for it to the authors.

Comments on the Quality of English Language

Minor editing of English language required and mostly pointed out in my revision

Reviewer 2 Report

Comments and Suggestions for Authors

Kim et al report the results of a laboratory toxicity study on the knockdown and lethal effects of transfluthrin on Aedes aegypti.

The results reveal that residue drying time is important as the compound losses efficacy over time.

I have some concerns that need to be addressed before the manuscript would be suitable for publication.

I have written suggestions and numbered points on a scanned copy of the manuscript.

Numbered points (see scanned file)

1. Indicate what these percentages refer to in the Simple Summary.

2. Indicate the name of the Ae. aegypti strain.

3a. What type of blood was used to feed female mosquitoes?

3b. What were the light conditions during the drying process? What type of light? What intensity?

Section 2.4: Did you use controls to estimate natural mortality of mosquitoes in the absence of pyrethroid?

4. Mann Whitney U statistics should be given with sample sizes (n1, n2). I was confused as zero values of U are only significant for small sample sizes.

5. Do you mean that the mortality responses to different TFT concentrations did not vary significantly when tested at 24 h compared to the 1 h responses. Suggest you reword for clarity.

6. What about control KD and mortality responses in Table 1? Were controls performed?

7. Should be written as Supplemental Table S1. Correct this throughout the manuscript. (Also with Tables S2 and S3.

8. The use of mixed effects models is not mention in Section 2.5.

9. Were controls performed at each time point in Table 2?

Section 3.4: It clear from Figure 3 that not all the LC and DCs were positively correlated with drying time.

10. You need to make it clear that none of these correlations were significant.

11. Fig 3. These slopes appeared to be wrong to me (DCs and LC99) – please check.

The section labeled Discussion contains a generic text that should be deleted.

12. The Discussion is rather long for such a simple set of results. Please reduce it by 20%.

The Tables in the Supplemental material should be labeled S1, S2 and S3.

References: the references contain some errors and typos (I only checked the first page)

Comments on the Quality of English Language

Minor editing.

Author Response

Thank you very much for your time for reviewing this manuscript. Please see the attachment.

Round 2

Reviewer 2 Report

Comments and Suggestions for Authors

The authors have markedly improved their manuscript. I have just a few points that need attention before publication.

1. L199, 200, 202: P values of zero should be changed to the conventional P <0.001

2. section 3.1 suggests to me that the authors have used Welch's t test (for unequal variances). This procedure is correct, but is not mentioned in the statistical analysis section 2.5.

3. Table 3. There is significant deviation from the Probit model in the 12 h sample (Chi2 = 10.4). The authors should mention why this is and the implications for the accuracy of the LC estimates.

4. Supplemental Table S2 could be improved by adjusting columns and altering text size to improve readability.

Comments on the Quality of English Language

Minor editing.
